# HIV Prevalence among Injury Patients Compared to Other High-Risk Groups in Tanzania

Loren K. Barcenas [1,†], Anna Tupetz [1,†], Shay Behrens [2], Arthi S. Kozhumam [3], Eleanor Strand [1], Megan von Isenburg [3], Philoteus A. Sakasaka [4], Matthew P. Rubach [3,5,6], Joao Ricardo Nickenig Vissoci [1,3], Lawrence P. Park [3,7], Janet Prvu Bettger [3,8,9], Blandina T. Mmbaga [3,4,10,11] and Catherine A. Staton [1,3,*]

1   Department of Emergency Medicine, School of Medicine, Duke University, Durham, NC 27705, USA
2   School of Medicine, Duke University, Durham, NC 27705, USA
3   Duke Global Health Institute, Duke University, Durham, NC 27705, USA
4   Kilimanjaro Clinical Research Institute, Moshi M8JG+2X7, Tanzania
5   Division of Infectious Diseases and International Health, Department of Medicine, School of Medicine, Duke University, Durham, NC 27705, USA
6   Programme for Emerging Infectious Diseases, Duke-National University of Singapore Medical School, Singapore 169857, Singapore
7   Infectious Diseases, Department of Medicine, Duke University, Durham, NC 27705, USA
8   Duke-Margolis Center for Health Policy, School of Medicine, Duke University, Durham, NC 27705, USA
9   Duke Clinical Research Institute, Duke University, Durham, NC 27705, USA
10  Kilimanjaro Christian Medical Centre, Moshi M8JG+2X7, Tanzania
11  Faculty of Medicine, Kilimanjaro Christian Medical University College, Tumaini University, Moshi M8JG+2X7, Tanzania
*   Correspondence: catherine.staton@duke.edu; Tel.: +1-919-684-5537
†   Designated first co-authors.

**Abstract:** Sixty-eight percent of persons infected with HIV live in Africa, but as few as 67% of those know their infection status. The emergency department (ED) might be a critical access point to HIV testing. This study sought to measure and compare HIV prevalence in an ED injury population with other clinical and nonclinical populations across Tanzania. Adults (≥18 years) presenting to Kilimanjaro Christian Medical Center ED with acute injury of any severity were enrolled in a trauma registry. A systematic review and meta-analysis was conducted to compare HIV prevalence in the trauma registry with other population groups. Further, 759 injury patients were enrolled in the registry; 78.6% were men and 68.2% consented to HIV counseling and testing. The HIV prevalence was 5.02% (tested), 6.25% (self-report), and 5.31% (both). The systematic review identified 79 eligible studies reporting HIV prevalence (tested) in 33 clinical and 12 nonclinical population groups. Notable groups included ED injury patients (3.53%, 95% CI), multiple injury patients (10.67%, 95% CI), and people who inject drugs (17.43%, 95% CI). These findings suggest that ED injury patients might be at higher HIV risk compared to the general population, and the ED is a potential avenue to increasing HIV testing among young adults, particularly men.

**Keywords:** HIV; AIDS; prevalence; injury; emergency department; Tanzania; Africa

## 1. Introduction

In 2014, the Joint United Nations Program on HIV/AIDS (UNAIDS) released the 90-90-90 targets for 2020, which, if reached, would have ended the pandemic by 2030 [1]. The 90-90-90 targets were as follows:

"By 2020, 90% of all people living with HIV will know their HIV status. By 2020, 90% of all people with diagnosed HIV infection will receive sustained antiretroviral therapy. By 2020, 90% of all people receiving antiretroviral therapy will have viral suppression" [1].

Unfortunately, these 90-90-90 targets were not to be reached. As a result, UNAIDS proposed an adjusted course of action, the Global AIDS Strategy 2021–2026, with an explicit

focus on reducing "disparities in access, HIV infections, and AIDS-related deaths" to reach 95-95-95 targets by 2025 [2].

Currently, the global burden of HIV/AIDS rests predominantly on Africa, where over two-thirds of the 38 million persons infected with HIV reside [3]. It is estimated that as few as 67% know their infection status [4]. To meet the new 95-95-95 targets, expanded testing strategies and access points are needed. Young adults, while 15% of the population, accounted for 28% of new HIV infections in 2019 [2]. Young adults tend to intersect with the health system at emergency departments (ED) given that unintentional injuries are the leading cause of death and disability for this age group [5].

Tanzania, a low- and middle-income country (LMIC) in East Africa, exhibits a high HIV prevalence and burden of injury. In 2019, the HIV prevalence was 4.8% [6], much higher than the continental average of 3.7% and six times the global average of 0.8% [7]. Injuries in Tanzania, primarily traffic- and violence-related, are responsible for over 10% of ED patients, the majority of whom are young adults [8–12]. Moreover, persons infected with HIV are at higher risk for injury-related mortality and morbidity than the general population [13].

A rapidly urbanizing nation, Tanzania is expected to see an increase in injuries due to growing motorcycle transport. To minimize the burden of injury and move toward the 95-95-95 targets, it is critical to identify access points for expanding HIV testing and reaching high-risk populations. This approach aligns with the national-provider-initiated HIV testing and counseling services (PITC) initiative launched in 2007. PITC aims to integrate HIV testing with standard care procedures in order to identify persons infected with HIV who happen to be seeking medical care for unrelated reasons (e.g., injury in the ED) but could be at increased risk for HIV/AIDS due to other high-risk behaviors [14].

The objective of this study was thus to (1) determine the prevalence of HIV among northern Tanzanian ED injury patients via a prospective cohort study, and (2) compare this ED population to the clinical and nonclinical population groups identified via a systematic review and meta-analysis. We hypothesize that ED injury patients will have a higher prevalence of HIV than the general population in Tanzania and thus provide evidence for the ED as an access point for HIV testing.

## 2. Materials and Methods

This study uses two methodological approaches. First, we conducted a prospective cohort study to determine the HIV prevalence of ED injury patients in northern Tanzania, following the Strengthening the Reporting of Observational Studies in Epidemiology (STROBE) guidelines [15]. STROBE guidelines ensure transparent reporting and are the standard for observational studies.

Second, we conducted a systematic review to identify and compare our injury patient HIV prevalence estimate with the current literature, following the Preferred Reported Items for Systematic Reviews and Meta-Analyses (PRISMA) guidelines [16]. A systematic review approach, as opposed to a scoping review or a literature review, was selected given the aim to evaluate prevalence through a meta-analysis if possible. PRISMA guidelines, also intended to ensure transparent reporting, are particularly suited to systematic reviews with an evaluation objective (e.g., prevalence, etiology, diagnosis).

Given the two methodological approaches, the methods and results are divided into two sections. The prospective cohort study is discussed first, followed by the systematic review and meta-analysis. At the end of the results section, we integrate the ED prevalence estimate with the estimates of population groups identified in the systematic review. The discussion expands on this integration and draws overall conclusions.

### 2.1. Prospective Cohort Study

2.1.1. Setting

The study setting was Kilimanjaro Christian Medical Center (KCMC) in Moshi, Tanzania. KCMC is located in a semi-urban area and serves as a referral hospital for 15 million

people across northern Tanzania [17]. Over 10% of patients at KCMC are impacted by injury. The injury patient population tends to be young men injured by road traffic injury and/or assault [18,19].

2.1.2. Procedures

All adults (≥18 years) presenting with an acute injury of any severity at the KCMC ED were enrolled in a trauma registry. Data were collected from 18 April 2018 to 6 September 2019. All enrolled patients were offered HIV testing and counseling.

HIV status was determined using the Tanzania National HIV Rapid Testing Algorithm for Persons Aged 18 Months or Older [20]. The algorithm stipulates testing with approved HIV rapid antibody tests, SD Bioline HIV-1/2 Rapid 3.0 (Seoul, South Korea), and/or Unigold HIV-1/2 rapid test (Trinity Biotech, Bray, Ireland). If the initial rapid antibody test is positive, a second rapid antibody test is used to confirm. Discordant results are settled by an ELISA test [21]. In this study, the primary screening test was the SD Bioline HIV-1/2, and the Unigold test was used for confirmation.

2.1.3. Descriptive Statistics and Variables

Patient demographics, injury characteristics, and HIV status are described. Categorical variables are presented as frequencies (percentages), and continuous variables are presented as medians and quartiles. Alcohol use within six hours prior to injury was determined by self-reported alcohol use, breathalyzer test upon ED arrival, or clinical examination.

Mechanism of injury was categorized as Road Traffic Injury (RTI), assault, or other. The "other" category consisted of mechanisms of injury reported as "other", "unknown", "drowning", and those with missing data.

The Kampala Trauma Score (KTS II) was used to measure injury severity and has been validated in numerous LMIC settings [22,23]. The score uses age, systolic blood pressure, respiratory rate, neurological status, and a score calculated for the number of serious injuries [24]. We categorized KTS II scores into mild (9–10), moderate (7–8), and severe (0–6).

Two HIV prevalence estimates were calculated: (1) among those who tested positive, and (2) among those who self-reported positive and tested positive.

*2.2. Systematic Review*

2.2.1. Eligibility Criteria

Target articles were peer-reviewed, quantitative studies estimating tested HIV prevalence in different adult population groups across Tanzania. Publication date was limited to after 2015 to reflect progress toward the UNAIDS 90-90-90 targets and recent epidemiology in Tanzania, including national prevalence data [25].

The exclusion criteria were abstract only; fact sheets; systematic reviews; and studies that obtained HIV status via self-report, did not describe data collection methods, and failed to report sample size. In addition, studies that used secondary national or census data as well as those who reported prevalence of the general population were excluded to avoid duplicating HIV prevalence already represented in the primary literature.

The search was performed in February 2022. Language of publication was limited to English.

2.2.2. Information Sources and Search

We searched the electronic databases Pubmed, Embase, Scopus, and African Index Medicus. All articles that fit the eligibility criteria in their title and abstract were included for full-text screening. The search strategy was built with terms related to HIV, AIDS, prevalence, incidence, epidemiology, and Tanzania (Appendix A, Table **??**).

### 2.2.3. Study Selection

We identified 1685 unique articles published up to February 2022 (Figure 1). Two independent researchers (E.S., A.S.K.) assessed eligibility and disagreements were resolved by study author A.T. Abstracts without sufficient information were excluded from full-text screening. COVIDENCE software was used to facilitate screening [26].

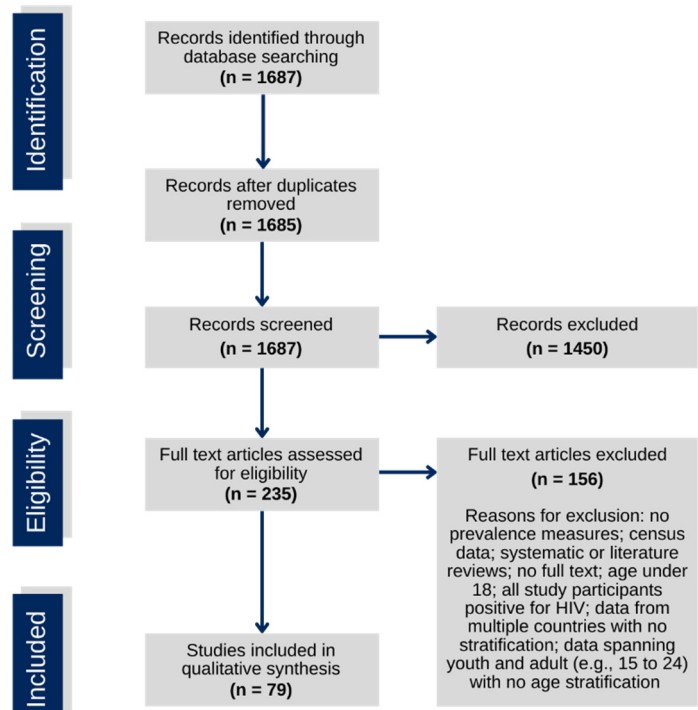

**Figure 1.** PRISMA flow diagram ('n' indicates number of articles).

### 2.2.4. Quality of Included Studies

Included studies had various study designs. We used a combination of the NIH Quality Assessment Tool for Observational Cohort and Cross-Sectional Studies [27] and Critical Appraisal Skills Program (CASP) for Case Control Study [28]. Our quality assessment guideline consisted of eight Yes/No questions extracted from the above tools (Appendix B). Studies were reviewed according to the guideline in duplicate by two independent researchers (E.S., A.S.K.). Studies with missing item(s) on the guideline were considered to have a risk of bias. No studies were excluded based on quality assessment.

### 2.2.5. Data Extraction

Two independent researchers (E.S., A.S.K.) extracted data in duplicate and disagreements were resolved via discussion. Data extraction included first author last name; year of publication; study objective; HIV prevalence as a primary or secondary outcome measure; study design; study setting; data collection method; year of data collection; sample size; sample size tested for HIV; study population characteristics; main results; and the author interpretation.

### 2.2.6. Data Analysis

To standardize our reporting, HIV prevalence was estimated for all studies via dividing the number of HIV-positive cases by the overall number of cases with known HIV status. Studies were categorized as clinical or nonclinical based on study population.

Once studies were stratified by population group, a meta-analysis was conducted to estimate pooled HIV prevalence for population groups with multiple studies. A random-effects model was chosen to account for between-study variance and allow for increased

generalizability to different scenarios and populations [29]. Analyses were performed via the DerSimonian–Laird estimation method [30].

Proportions were calculated with a generalized linear mixed methods approach. Heterogeneity was examined via Cochrane's Q (considers *p*-values under 0.05 as indicators of heterogeneity), H, and I statistics [31]. High $I^2$ values indicate high heterogeneity; categories include low (25%), moderate (45%), and high (75%) [32].

Between-study variance (tau) was estimated with a maximum-likelihood estimator. Study bias was evaluated through descriptive statistics and exploratory graphical analysis via funnel and forest plots.

All analyses were performed using R software [33], specifically the meta and metafor packages. HIV prevalence in our injury patient sample was compared to populations identified in the systematic review via graphical depiction.

## 3. Results

The results are reported first from the prospective cohort study, then the systematic review and meta-analysis. A comparison at the end situates the ED injury patient population prevalence within those identified in the systematic review.

### 3.1. Prospective Cohort Study

In the trauma registry, 759 injury patients were enrolled (Figure 2), and 518 (68.2%) agreed to HIV testing and counseling. Of these, 26 (4.1%) were HIV-positive; 17 of the 26 self-reported a positive status; the remaining nine were newly diagnosed. Of those who tested negative (492 out of 518), 302 had self-reported negative status from previous testing.

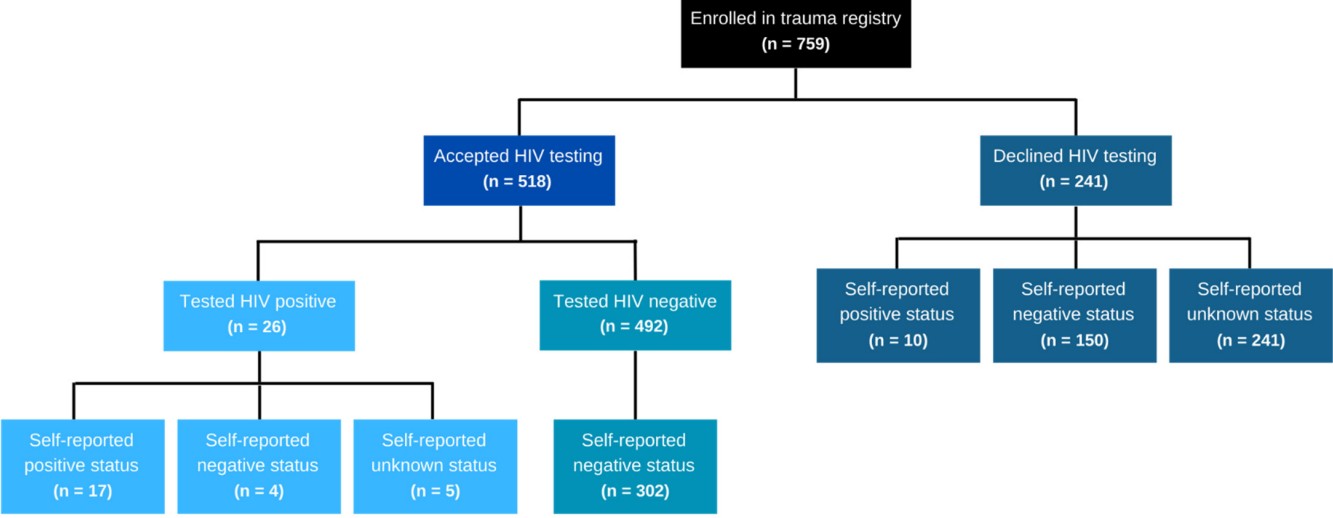

**Figure 2.** ED injury patient population stratified by HIV status ('n' indicates number of patients).

Among the 241 patients (31.8%) who declined HIV testing and counseling, 10 self-reported a positive status, 150 self-reported a negative status, and 81 self-reported an unknown status. Further, 64 of the 81 (80%) with unknown status were men. In total, both self-report and tested, 36 patients were HIV-positive, 642 were HIV-negative, and 81 were unknown. The HIV prevalence was 5.02% (tested), 6.25% (self-report), and 5.31% (both).

Further, 593 of the 759 (78.6%) were men, the median age was 34 (IQR 26–47) (Table 1), and 157 of the 759 (20.7%) either reported consuming alcohol within six hours prior to injury or tested positive on a breathalyzer upon admission to the ED. Patients with a positive or unknown HIV status exhibited higher alcohol use than those with negative status.

**Table 1.** ED injury patient population demographics stratified by HIV status ('n' indicates number of patients).

| Characteristic | Total Sample (n = 759) | HIV-Positive (Both) (n = 36) | HIV-Negative (Tested) (n = 492) | HIV-Negative (Self-Report) (n = 150) | Unknown Status (n = 81) |
|---|---|---|---|---|---|
| **Age** Median (IQR) | 34 (26–47) | 39 (31.8–50) | 33 (26–46) | 35 (28–45) | 32 (24–49) |
| **Male** n (%) [missing] | 593 (78.6%) [5] | 20 (55.6%) [0] | 392 (80.2%) [3] | 117 (78.5%) [1] | 64 (80%) [1] |
| **Alcohol use within 6 h of injury** n (%) | 157 (20.7%) | 10 (27.8%) | 96 (19.5%) | 32 (21.3%) | 19 (23.5%) |
| **Years of education** Median (IQR) | 7 (7–11) | 7 (7–8.5) | 7 (7–11) | 7 (7–11) | 7 (7–11) |
| **Employment** n (%) [missing] | [3] | [0] | [2] | [0] | [1] |
| Student | 24 (3.2%) | 0 | 14 (2.8%) | 5 (3.3%) | 5 (6.2%) |
| Unemployed | 27 (3.6%) | 1 (2.8%) | 15 (3%) | 5 (3.3%) | 6 (7.4%) |
| Professional | 84 (11.1%) | 3 (8.3%) | 56 (11.4%) | 21 (14%) | 4 (4.9%) |
| Skilled employment | 117 (15.4%) | 5 (13.9%) | 78 (15.9%) | 23 (15.3%) | 11 (13.6%) |
| Self-employed | 261 (34.4%) | 17 (47.2%) | 159 (32.3%) | 54 (36%) | 31 (38.3%) |
| Farmer | 211 (27.8%) | 8 (22.2%) | 148 (30.1%) | 36 (24%) | 19 (23.5%) |
| Other | 32 (4.2%) | 2 (5.6%) | 20 (4.1%) | 6 (4%) | 4 (4.9%) |
| **Prior testing** n (%) | 485 (63.9%) | 31 (86.1%) | 304 (61.8%) | 150 (100%) | 0 |
| **Mechanism of injury** n (%) [missing] | [1] | [0] | [0] | [0] | [1] |
| Road traffic injury | 478 (63%) | 20 (55.6%) | 315 (64%) | 89 (59.3%) | 54 (66.7%) |
| Assault | 100 (13.2%) | 5 (13.9%) | 62 (12.6%) | 23 (15.3%) | 10 (12.3%) |
| Other | 180 (23.7%) | 11 (30.6%) | 115 (23.4%) | 38 (25.3%) | 16 (19.8%) |
| **KTS II** n (%) [missing] | [99] | [9] | [46] | [32] | [12] |
| Mild (9, 10) | 341 (44.9%) | 16 (44.4%) | 229 (46.5%) | 65 (43.3%) | 31 (38.3%) |
| Moderate (7, 8) | 307 (40.4%) | 10 (27.8%) | 211 (42.9%) | 51 (34%) | 35 (43.2%) |
| Severe (0–6) | 12 (1.6%) | 1 (2.8%) | 6 (1.2%) | 2 (1.3%) | 3 (3.7%) |
| **Marital status** n (%) [missing] | [1] | [0] | [1] | [0] | [0] |
| Single | 266 (35%) | 12 (33.3%) | 169 (34.3%) | 49 (32.7%) | 36 (44.4%) |
| Married | 404 (53.2%) | 15 (41.7%) | 268 (54.5%) | 80 (53.3%) | 41 (50.6%) |
| Partner, not married | 15 (2%) | 1 (2.8%) | 13 (2.6%) | 1 (0.7%) | 0 |
| Widow/widower | 32 (4.2%) | 3 (8.3%) | 18 (3.7%) | 9 (6%) | 2 (2.5%) |
| Separated | 41 (5.4%) | 5 (13.9%) | 23 (4.7%) | 11 (7.3%) | 2 (2.5%) |

Road traffic injury (63%) and assault (13.2%) were the most common mechanisms of injury. Further, 83% of assault patients were men; 660 (87%) patients had sufficient data to calculate the KTS II. The majority (44.9%) had mild injury. The distribution of injury mechanisms across HIV status was similar. HIV prevalence was highest among patients with 'other' injury (6.1%), followed by assault (5.0%) and road traffic injury (4.2%).

Further, 485 of the 759 (63.9%) self-reported previous HIV testing. Of the 26 who tested positive, information on prior knowledge of status was missing in five. Four patients self-reported negative and tested positive. This group had the highest median viral load (7615; IQR 39–103364) compared to those who self-reported positive (39; IQR 0–141). The CD4 count was similar between patients aware of their positive status (403; IQR 258.5–652), newly diagnosed (394; IQR 267–673), and self-reported negative (424.5; IQR 384–510.2).

Most patients who self-reported positive (76.4%) were on antiretroviral (ARV) treatment. Two patients who self-reported negative but tested positive indicated they were

already on ARV. One patient reported tuberculosis as an AIDS-related infection. Of the 26 patients infected with HIV, seven reported HIV-positive family members. Of those, three reported more than one HIV-positive family member (Table 2).

**Table 2.** Testing characteristics of ED injury patients with HIV-positive status ('n' indicates number of patients).

| | | Tested HIV-Positive (n = 26) | Self-Reported and Tested HIV-Positive (n = 17) | Self-Reported HIV-Negative, Tested HIV-Positive (n = 4) |
|---|---|---|---|---|
| **HIV test results from ED visit** | **Viral load (copies/mL)** Median (IQR) | 54.5 (9.8–12,959.5) | 39 (0–141) | 7615 (39–103,364) |
| | **CD4 count** Median (IQR) [missing] | 403 (258.5–652) [2] | 394 (267–673) | 424.5 (384–510.2) |
| **Self-reported prior HIV testing and/or treatment** | **On ARV** n (%) [missing] | 15 (57.7%) | 13 (76.4%) [3] | 2 (50%) [1] |
| | **AIDS-related infections** n (%) | 1 (3.8%) | 0 | 0 |
| | **HIV-positive family** n (%) [missing] | 7 (26.9%) | 5 (29.4%) | 1 (25%) [1] |

*3.2. Systematic Review*

3.2.1. Study Characteristics

Although the 79 included articles were published between 2015 and 2022, data collection for some long-term cohort studies started in 2004 (Table 3). The sample size of individuals with known HIV status ranged from 45 [34] to 133,695 [35]. Further, 42 studies assessed HIV status and measured HIV prevalence as a primary outcome measure [35–76]. Studies either had no missing items (n = 28) or one missing item (n = 51) on the quality assessment guideline (Appendix C, Table A2). The missing item for these 51 studies was sample size calculations and rationale.

3.2.2. Narrative Summary of Results

Of the 79 included studies, there were 45 population groups, 33 clinical and 12 nonclinical. The most commonly investigated population groups were TB-related (n = 13), women in various phases of pregnancy (n = 9), people who inject drugs (n = 8), men who have sex with men (n = 6), and women tested for or with cervical cancer (n = 6). Other population groups included the general population of different age groups and genders, specific professions, and patients seeking care for other health concerns (e.g., head and neck cancer, Fournier's gangrene).

The reported individual HIV prevalence ranged from 1.7% in adults over age 50 [75] to 66% in presumptive pulmonary TB patients [77]. Adults over age 50 also had the lowest HIV pooled prevalence (3.23%) [65,66,75,78]. Confirmed TB patients were found to have the highest pooled prevalence (35.06%) among all included studies [42,44,79–81] (Table 3).

For the 33 clinical population groups, the overall random-effects HIV pooled prevalence was 13.36% (95% CI, 10.28%–17.37%). Individual HIV prevalence ranged from 1.9% in blood donors [82] to 66% in presumptive pulmonary TB patients [77]. Pooled prevalence was lowest in blood donors (2.73%) and highest in confirmed TB patients (35.06%) (Table 3).

**Table 3.** (a) Study characteristics and HIV prevalence for clinical study population groups (n = 33) ('n' indicates number of population groups). (b) Study characteristics and HIV prevalence for nonclinical population groups (n = 12) ('n' indicates number of population groups).

| (a) | | | | | | | |
|---|---|---|---|---|---|---|---|
| **Population Group** | **Study** | **Year of Data Collection** | **Geographic Region** | **Sample Size HIV Tested** | **Prevalence (95% CI)** | **Pooled-Prevalence (95% CI)** | **Model Heterogeneity** |
| **Pregnant women** | Gamell, 2017 [43]<br>Konje, 2018 [53]<br>Chibwe, 2019 [83]<br>Ng'wamkai, 2019 [67] | 2014–2015<br>2016–2017<br>2017<br>2018 | Kilombero district<br>Geita district<br>Mwanza<br>Mwanza | 1548<br>1426<br>291<br>499 | 3.10%<br>3.88%<br>3.40%<br>5.01% | 3.75%<br>[3.05%–4.62%] | $I^2 = 27\%$<br>$\tau^2 = 0.0141$<br>$p = 0.25$ |
| **Pregnant women attending antenatal care clinics (ANC)** | Manyahi, 2017 [58] | 2014 | Temeke municipality | 249 | 17.20% | N/A | N/A |
| **Pregnant women at delivery** | Lawi, 2015 [54] | 2012 | Mwanza | 408 | 7.20% | N/A | N/A |
| **Women who delivered within the last 2 years** | Adinan, 2019 [36] | 2017 | Geita region | 767 | 3.39% | N/A | N/A |
| **Newly delivered mothers** | Nungu, 2019 [71] | 2015–2016 | Wanging'ombe and Njombe districts | 668 | 4.04% | N/A | N/A |
| **Women screened for cervical cancer** | Chambuso, 2016 [39]<br>Mchome, 2020 [84]<br>Chinn, 2021 [85]<br>Katanga, 2021 [86] | unknown<br>2015–2017<br>2018<br>unknown | Morogoro<br>Dar es Salaam<br>Mwanza<br>unknown | 517<br>4043<br>824<br>3643 | 21.30%<br>17.80%<br>8.01%<br>17.90% | 17.54%<br>[16.77%–18.35%] | $I^2 = 94\%$<br>$\tau^2 = 0.1742$<br>$p < 0.01$ |
| **Women with cervical cancer** | Lovgren, 2016 [56]<br>Khamis, 2021 [87] | 2007–2011<br>2012 | Dar es Salaam<br>Dar es Salaam | 143<br>202 | 38.46%<br>9.40% | 19.27%<br>[4.85%–76.58%] | $I^2 = 97\%$<br>$\tau^2 = 0.9622$<br>$p < 0.01$ |
| **Men undergoing voluntary circumcision** | Bazant, 2016 [88] | 2014–2015 | Iringa, Njombe, Tabora | 665 | 11.90% | N/A | N/A |
| **Febrile adult patients** | Boillat-Blanco, 2018 [89] | 2013–2014 | Dar es Salaam | 519 | 25.00% | N/A | N/A |
| **Patients with Fournier's gangrene** | Chalya, 2015 [90] | 2006–2014 | Bugando | 80 | 11.30% | N/A | N/A |
| **Patients who were managed for domestic-violence-related trauma** | Chalya, 2015 [91] | 2009–2014 | Bugando | 324 | 7.10% | N/A | N/A |
| **Hospital patients** | Kilale, 2016 [92] | 2010–2012 | Arusha municipality | 664 | 24.00% | N/A | N/A |
| **Patients in outpatient department clinics** | Cham, 2019 [35] | 2014–2017 | Bukoba | 133695 | 4.20% | N/A | N/A |
| **Multiple injury patients** | Issa, 2018 [93] | 2013 | Bugando | 150 | 10.70% | N/A | N/A |
| **ED patients with injury** | Hyuha, 2021 [45] | 2019–2020 | Dar es Salaam | 255 | 3.50% | N/A | N/A |
| **Confirmed TB patients** | Denti, 2015 [79]<br>Gunda, 2017 [44]<br>Friis, 2018 [42]<br>Kidenya, 2018 [80]<br>Mhimbira, 2019 [81] | 2010–2011<br>2016–2017<br>2006–2009<br>2014–2015<br>2013–2015 | Mwanza<br>Sengerema district<br>Mwanza<br>Mwanza<br>Dar es Salaam | 100<br>156<br>1605<br>78<br>794 | 50.00%<br>35.26%<br>41.43%<br>34.60%<br>21.16% | 35.06%<br>[26.22%–46.89%] | $I^2 = 96\%$<br>$\tau^2 = 0.1001$<br>$p < 0.01$ |
| **Patients on treatment for TB** | Munseri, 2019 [94] | 2016–2017 | Dar Es Salaam | 660 | 31.12% | N/A | N/A |
| **Suspected TB patients** | Hoza, 2016 [95]<br>Reither, 2015 [96] | 2012–2013<br>2012 | Ngamiani, Muheza, Bombo, Makorora<br>Bagamoyo district | 372<br>480 | 14.20%<br>42.49% | 24.76%<br>[8.48%–72.25%] | $I^2 = 98\%$<br>$\tau^2 = 0.5878$<br>$p < 0.01$ |
| **Presumptive pulmonary TB patients** | Mhimbira, 2015 [77] | unknown | Bagomoyo district | 143 | 66% | N/A | N/A |
| **Patients with bacteriologically confirmed pulmonary TB** | Senkoro, 2016 [97] | 2011–2012 | Dar es Salaam | 151 | 7.95% | N/A | N/A |
| **Patients who completed 20–24 weeks of TB treatment** | Manji, 2016 [98] | 2014 | Temeke municipality | 501 | 30.30% | N/A | N/A |
| **Patients treated with levofloxacin as part of MDR-TB regimen** | Mohamed, 2021 [34] | 2019 | Northern Tanzania | 45 | 35.56% | N/A | N/A |

**Table 3.** *Cont.*

| | | | | | | | |
|---|---|---|---|---|---|---|---|
| Adults who received TB treatment within 2 years | Mpagama, 2021 [99] | unknown | Kilimanjaro | 219 | 15.98% | N/A | N/A |
| Head and neck cancer patients | Gilyoma, 2015 [100] | 2009–2013 | Mwanza | 346 | 7.20% | N/A | N/A |
| PWID attending methadone clinics | Lambdin, 2017 [101] | 2011–2013 | Dar es Salaam | 630 | 40% | 22.74% [7.36%–70.26%] | $I^2 = 98\%$ $\tau^2 = 0.6480$ $p < 0.01$ |
| | Kilonzo, 2021 [52] | 2019–2020 | Mwanza | 253 | 12.80% | | |
| HCV-seropositive patients enrolled in the local opioid substitution treatment center | Mohamed, 2017 [102] | 2015 | Dar es Salaam | 116 | 43.97% | N/A | N/A |
| Patients with a neurological disorder admitted to the medical ward | Laizer, 2019 [103] | 2007–2008 | Moshi (KCMC) | 337 | 20.50% | N/A | N/A |
| Blood donors | Lidenge, 2020 [104] | 2019 | Dar es Salaam | 504 | 4.20% | 2.73% [1.27%–5.89%] | $I^2 = 93\%$ $\tau^2 = 0.2851$ $p < 0.01$ |
| | Mremi, 2021 [82] | 2017–2019 | Kilimanjaro | 101616 | 1.90% | | |
| Patients with first stroke | Matuja, 2020 [105] | 2018–2019 | Dar es Salaam | 369 | 11.92% | N/A | N/A |
| Patients with chronic kidney disease (CKD) | Meremo, 2018 [106] | 2013–2015 | Dodoma | 792 | 4.80% | N/A | N/A |
| Women with and without *S. mansoni* infection | Mishra, 2019 [107] | unknown | Kisesa, Lumeji, Welamasonga, Kayenze | 97 | 10.00% | N/A | N/A |
| Patients with surgical acute abdomen | Sravanam, 2018 [74] | 2016 | Mwanza | 106 | 14.20% | N/A | N/A |
| VCT clients, ANC attendees, blood donors, and CTC patients | Urio, 2015 [76] | 2011–2012 | unknown | 596 | 35.10% | N/A | N/A |

**(b)**

| Population group | Study | Year of data collection | Geographic region | Sample size HIV tested | Prevalence (95% CI) | Pooled-prevalence (95% CI) | Model heterogeneity |
|---|---|---|---|---|---|---|---|
| Cohabiting couples | Ngilangwa, 2015 [68] | 2005–2007 | Kilimanjaro, Arusha | 2666 | 13.02% | N/A | N/A |
| Married and cohabitating heterosexual adults | Mtenga, 2015 [64] | 2013 | Ifakara | 3737 | 6.69% | N/A | N/A |
| Adults over 50 | Senkoro, 2016 [78] | 2011–2012 | Dar es Salaam | 6302 | 5.05% | 3.23% [1.65%–6.31%] | $I^2 = 89\%$ $\tau^2 = 0.4193$ $p < 0.01$ |
| | Mtowa, 2017 [65] | 2012–2013 | Ifakara | 1643 | 6.03% | | |
| | Swai, 2017 [75] | 2015 | Rombo district | 588 | 1.70% | | |
| | Muiruri, 2019 [66] | 2015 | Rombo district | 600 | 1.70% | | |
| Adult women | Faber, 2017 [41] | 2008–2009 | Dar es Salaam, Pwani, Mwanza, | 3424 | 10.19% | 8.48% [4.91%–14.67%] | $I^2 = 95\%$ $\tau^2 = 0.3037$ $p < 0.01$ |
| | Hjort, 2019 [108] | 2014–2015 | and Mtwara Korogwe, Tanga | 952 | 3.57% | | |
| | Baldur-Felskov, 2019 [109] | 2008–2009 | Dar es Salaam, Pwani, Mwanza, and Mtwara; | 3339 | 10.00% | | |
| | Mchome, 2021 [60] | 2015–2016 | Dar es Salaam and Kilimanjaro | 2253 | 13.30% | | |
| Sexually active women | Safari, 2019 [110] | 2015–2016 | Magu district | 4052 | 8.09% | N/A | N/A |
| Female bar workers | Barnhart, 2019 [38] | 2017 | Kinondoni district | 56 | 7.10% | N/A | N/A |
| Adult men | Norris, 2017 [69] | 2004 | Dar es Salaam, Pwani, Tanga, | 158 | 8.86% | 9.27% [7.98%–10.78%] | $I^2 = 0\%$ $\tau^2 = 0$ $p = 0.85$ |
| | Olesen, 2017 [111] | 2009 | northern Tanzania Kilimanjaro | 1503 | 9.31% | | |
| Men living in rural Tanzania | Downs, 2017 [40] | 2014–2016 | Mwanza | 674 | 5.60% | N/A | N/A |
| Men who have sex with men (MSM) | Ahaneku, 2016 [37] | 2012–2013 | Dar es Salaam, Tanga | 176 | 25% | 14.36% [10.85%–19.01%] | $I^2 = 89\%$ $\tau^2 = 0.1115$ $p < 0.01$ |
| | Ishungisa, 2020 [46] | 2017 | Dar es Salaam | 777 | 12.30% | | |
| | Khatib, 2017 [51] | 2007; 2011 | Unguja, Zanzibar | 848 | 8.72% | | |
| | Mmbaga, 2017 [63] | 2015 | Dar es Salaam | 610 | 15.50% | | |
| | Mmbaga, 2020 [112] | 2014 | Dodoma | 409 | 17.36% | | |
| | Mizinduko, 2020 [61] | 2017 | Dar es Salaam | 777 | 12.36% | | |

**Table 3.** *Cont.*

| Male plantation residents | Norris 2017 [70] | 2004 | Northern Tanzania | 158 | 8.86% | N/A | N/A |
|---|---|---|---|---|---|---|---|
| **Fisherfolk** | Kapesa, 2018 [47] | 2017 | Selected Islands of Lake Victoria in Buchosa and Muleba districts | 456 | 14.00% | 12.18% [9.12%–16.27%] | $I^2 = 86\%$ $\tau^2 = 0.0561$ $p < 0.01$ |
| | Kapiga, 2021 [48] | 2015–2016 | Muleba, Sengerema, and Ukerewe along Lake Victoria | 1121 | 14.20% | | |
| | Panga, 2021 [72] | 2019 | Geita and Chato districts | 1048 | 9.06% | | |
| **People who inject drugs (PWID)** | Matiko, 2015 [59] | 2007; 2012 | Zanzibar | 907 | 16.09% | 15.82% [10.86%–23.06%] | $I^2 = 95\%$ $\tau^2 = 0.2109$ $p < 0.01$ |
| | Khatib, 2017 [50] | 2012 | Zanzibar | 408 | 11.30% | | |
| | Mmbaga, 2017 [62] | 2015 | Dar es Salaam | 610 | 15.50% | | |
| | Leyna, 2019 [55] | 2017 | Dar es Salaam | 611 | 8.35% | | |
| | Kawambwa, 2020 [49] | 2017 | Dar es Salaam, Sinza, Kinondoni, Kimara, Tandale, Msasani, Mbagala, Kunduchi, Temeke, and Tandika suburbs | 219 | 33.80% | | |
| | Minja, 2021 [113] | 2016–2017 | unknown | 897 | 18.84% | | |

Abbreviations: ANC (antenatal care); TB (tuberculosis); MDR-TB (multi-drug-resistant TB); PWID (persons who inject drugs); HCV (hepatitis C virus); VCT (voluntary testing and counseling); CTC (Muhimbili—Care and Treatment Center).

For the 12 nonclinical population groups included in the review, the random-effects HIV pooled prevalence was 9.79% (95% CI, 7.88–12.16%). Individual HIV prevalence ranged from 1.7% in adults over age 50 [75] to 33.80% in people who inject drugs [49]. Pooled prevalence was lowest in adults over age 50 (3.23%) [65,66,75,78] and highest in people who inject drugs (15.82%) [49,50,55,59,62,108].

### 3.3. Comparison of Findings

The HIV prevalence among our sample is between 5.02% (tested) and 6.25% (self-report), while the national average is 4.8% (Figures 3 and 4). Injury patients in our sample had a lower HIV prevalence than adult women; fisherfolk; cohabitating couples; men who have sex with men (MSM); people who inject drugs (PWID); adults who received TB treatment within two years; pregnant women attending antenatal care clinics; women screened for cervical cancer; patients with a neurological disorder admitted to medical wards; hospital patients; febrile adult patients; patients who completed 20–24 weeks of treatment; patients on treatment for TB; VCT clients, ANC attendees, blood donors, and CTC patients; patients treated with levofloxacin as part of a regimen for MDR-TB; suspected TB patients; PWID attending methadone clinics; confirmed TB patients; HCV-seropositive patients enrolled in the local opioid substitution treatment center; and presumptive pulmonary TB patients.

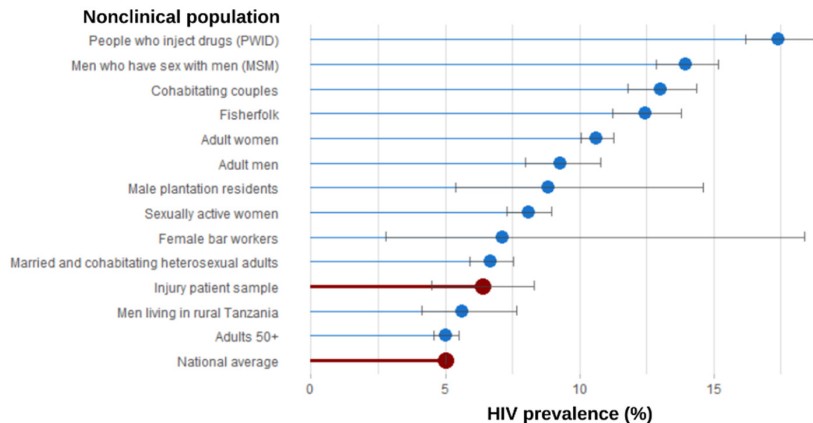

**Figure 3.** Comparing HIV prevalence estimates of nonclinical population groups identified via systematic review with the ED injury patient sample and national average.

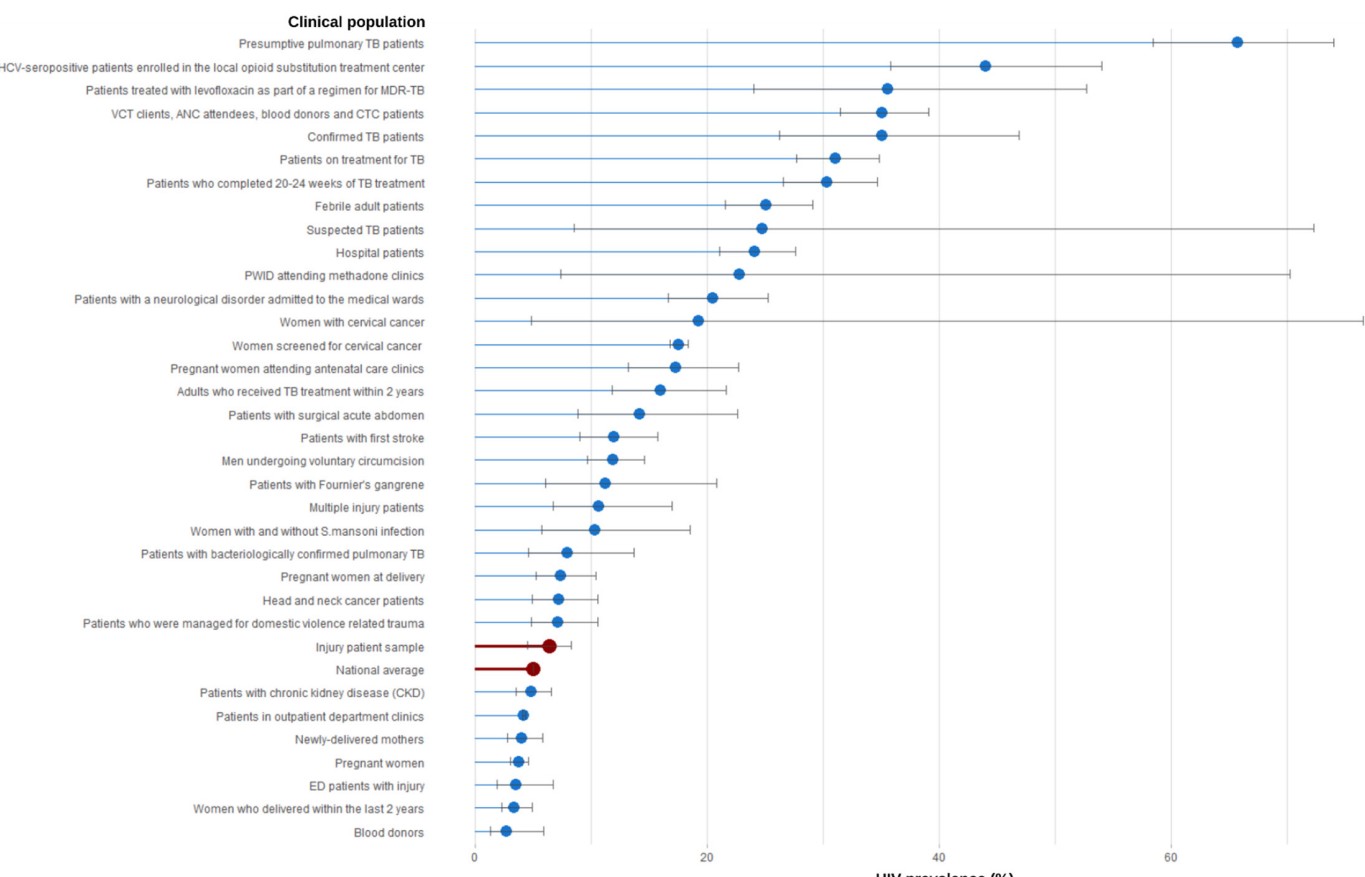

**Figure 4.** Comparing HIV prevalence estimates of clinical population groups identified via systematic review with ED injury patient sample and national average.

Other populations that had a range similar to our sample included married and cohabitating heterosexual adults; female bar workers; sexually active women; male plantation residents; adult men; patients who were managed for domestic-violence-related trauma; head and neck cancer patients; pregnant women at delivery; patients with bacteriologically confirmed pulmonary TB; women with and without *S. mansoni* infection; multiple injury patients; patients with Fournier's gangrene; men undergoing voluntary circumcision; patients with first stroke; patients with surgical acute abdomen; women with cervical cancer.

## 4. Discussion

To our knowledge, this study was the first to estimate HIV prevalence among acute ED injury patients and the first systematic review of HIV prevalence stratified by clinical and nonclinical population groups in Tanzania. Given that injury patients often present with increased risk behaviors, such as reckless driving and excessive alcohol use [114–117], we hypothesized that this extended to high-risk sexual behaviors and thus a higher HIV prevalence. Our results concur, suggesting that ED injury patients are a moderately high-risk population group with an HIV prevalence at or above the general population but below other well-known high-risk groups, such as PWID or sex workers and their partners [118].

In ED injury patients, the rate of alcohol use, from high to low, was as follows: persons infected with HIV who knew their status, persons who had never tested (i.e., unknown status), and persons who tested negative. The high rate of alcohol use in persons infected with HIV supports our hypothesized positive correlation between injury and high-risk behavior [114–117]. In 2014, a World Health Organization (WHO) report on injury and violence stated that increased risk behaviors, notably unsafe sexual practices, are a potential

consequence of physical injury—leading to a higher prevalence of HIV/AIDS and other sexually transmitted infections (STI) [119].

In Tanzania, men are at highest risk for injury, representing over 70% of ED injury patients [120]. Our findings were consistent with these demographics given that men accounted for 78.6% of our injury patient sample. In terms of HIV testing, a recent HIV/AIDS impact survey reported that 41% of men in Tanzania have never been tested for HIV [20]. Our injury patient sample was again consistent, with men representing 80% of those who had never been tested. The ED, therefore, is a nexus between a high-risk population—men, usually young adults, who engage in high-risk behaviors and have not tested for HIV—and the health care system.

Adding HIV testing to the standard of care for patients presenting with injury in EDs would move us toward the 95-95-95 targets, in particular 95% of persons knowing their status, and align with the national PITC approach [14]. Tanzania, furthermore, launched the Male Catch-Up Plan 2020, a social behavior change initiative to increase HIV testing among heterosexual men [121]. Integrating HIV testing with the ED would directly contribute to these two national, government-backed initiatives.

With an opt-out, as opposed to an opt-in, testing is recommended [122]. Opt-out protocols have been proven to increase HIV testing in EDs (25). In 2016, an ED in South Africa integrated HIV testing with its standard of care. There was a "high uptake of HIV testing (78.6%) among a predominantly male (58%) patient group who mostly presented with traumatic injuries (70.8%)" [123]. Of the men who tested positive for HIV, approximately two-thirds were previously unaware of their status [123]. The need for, and benefit of, HIV testing in the ED is clear.

Feasibility, moreover, has been proven. A study at an ED in Dar es Salaam, Tanzania evaluated the feasibility of integrating opt-out HIV testing with routine care for injury patients. The HIV prevalence was 5.6% (similar to our sample), and 75% of men accepted testing. Notably, it was reported that none of these patients would have been tested if not for integration—identifying a need to promote and enforce the PITC initiative [124].

The ED injury patient population is likely at high risk for HIV and understudied in Tanzania. With a minimum HIV prevalence at or above the national average and a maximum near double, it is critical to expand HIV testing to the ED. Future studies should further evaluate the feasibility of integrating opt-out HIV testing with routine care in EDs, in particular in connection with the PITC and Male Catch-Up Plan government initiatives, as well as barriers and facilitators to accessing, and accepting, HIV testing among the ED injury patient population.

## 5. Limitations

This study has methodological limitations that should be considered when interpreting the results. The prospective cohort study limitations are presented first, followed by the systematic review. As a whole, given that the analysis was limited to Tanzania, the results should not be generalized outside of Tanzania until further studies are conducted.

### 5.1. Prospective Cohort Study

While the sample size is 759, our hypotheses and inferences are based on 36 positive cases, which limits our discussion to trends rather than statistically significant associations. Our data are from one ED in a semi-urban area and might not be representative of the nation. This hospital, however, serves as a referral center for over 15 million people across northern Tanzania [17], and other studies (discussed above) have aligned with our findings. In the data, self-report HIV status might not be correct, but there were no major differences between tested and self-reported prevalence.

### 5.2. Systematic Review

All the included studies either had no missing or one missing item according to the quality assessment guidelines. Several studies had a subset of their sample tested for HIV.

This was accounted for in the meta-analysis, but there could be selection bias among those who chose to get tested. Some studies with eligible data were excluded because the data were not age-stratified (e.g., age 15 to 24). In addition, prevalence data in unpublished theses, non-indexed journals (common in LMICs), and gray literature would have been excluded. Peer-reviewed publication as an inclusion criterion was necessary, however, to ensure data of sufficient rigor for the meta-analysis.

## 6. Conclusions

This study sought to measure and compare HIV prevalence in the ED injury patient population with other clinical and nonclinical population groups across Tanzania. The prospective cohort study estimated HIV prevalence for ED injury patients and confirmed that emergency departments are a nexus between a high-risk population group—men, usually young adults, who engage in high-risk behaviors and have not tested for HIV—and the health care system. The systematic review and meta-analysis demonstrated ED injury patients are at moderate to high risk, with an HIV prevalence at or above the national average but below other high-risk groups. These findings suggest that injury patients might be at higher HIV risk compared to the general population. Integrating opt-out HIV testing with routine care in the ED is a potential avenue to increasing testing among young adults, particularly men, and moving toward the 95-95-95 targets.

**Author Contributions:** Conceptualization, S.B., A.T., J.R.N.V., B.T.M., J.P.B., C.A.S. and M.v.I.; methodology, L.K.B., A.T., A.S.K., P.A.S., M.P.R., J.R.N.V., L.P.P., J.P.B., C.A.S. and M.v.I.; formal analysis, L.K.B., A.T., S.B., A.S.K., P.A.S., M.P.R., J.R.N.V., E.S. and M.v.I.; investigation, L.K.B., A.T., S.B., A.S.K., P.A.S., M.P.R., J.R.N.V., E.S. and M.v.I.; writing—original draft preparation, L.K.B., A.T., S.B., A.S.K. and E.S.; writing—review and editing, L.K.B., A.T., P.A.S., M.P.R., J.R.N.V., L.P.P., J.P.B., B.T.M., C.A.S. and E.S.; visualization, E.S.; supervision, M.P.R., J.R.N.V., J.P.B., B.T.M. and C.A.S.; funding acquisition, C.A.S., J.P.B. and BTM. All authors have read and agreed to the published version of the manuscript.

**Funding:** Research reported in this publication was supported by the Fogarty International Center of the National Institute of Health under Award Number R21 TW0010456. The content is solely the responsibility of the authors and does not necessarily represent the official views of the National Institute of Health.

**Institutional Review Board Statement:** This study protocol was approved by the Duke University Medical Center Institutional Review Board under the IRB protocol number Pro00086496 and the Kilimanjaro Christian Medical Center Ethics Committee and the Tanzanian National Institute of Medical Research.

**Informed Consent Statement:** Written or verbal informed consent was obtained from all participants involved in the study.

**Data Availability Statement:** Detailed search strategy for systematic review attached as Appendix A to re-create the search and access included articles. All quantitative HIV data can be available upon request from Gwamaka William due to the sensitive nature of the data and the fact that it is under a data sharing agreement.

**Acknowledgments:** We would like to acknowledge our Kilimanjaro Christian Medical Center Emergency Department Research team without whom none of our projects would happen. For this manuscript, it was especially 'Mama' Amina Mlay, Sister Anna Mchaki, and Sister Eliza Ngowi who have always been dedicated to ensuring the best care for patients at KCMC. Your dedication and service to KCMC patients is exemplary. A special thanks to Ashley Philips for her guidance and editing of the manuscript.

**Conflicts of Interest:** The authors declare no conflict of interest. The funders had no role in the design of the study; in the collection, analyses, or interpretation of data; in the writing of the manuscript; or in the decision to publish the results.

## Abbreviations

| | |
|---|---|
| HIV | Human immunodeficiency virus |
| AIDS | Acquired immunodeficiency syndrome |
| ED | Emergency department |
| KCMC | Kilimanjaro Christian Medical Center |
| UNAIDS | Joint United Nations Program on HIV/AIDS |
| LMIC | Low- and middle-income country |
| PITC | Provider-initiated HIV testing and counseling services |
| STROBE | Strengthening the Reporting of Observational Studies in Epidemiology |
| PRISMA | Preferred Reporting Items for Systematic Reviews and Meta-Analyses |
| RTI | Road traffic injury |
| KTS II | Kampala Trauma Score II |
| CASP | Critical Appraisal Skills Program |
| TB | Tuberculosis |
| MSM | Men who have sex with men |
| WHO | World Health Organization |
| PWID | Persons who inject drugs |
| ANC | Antenatal care |
| MDR-TB | Multi-drug-resistant TB |
| HCV | Hepatitis C virus |
| VCT | Voluntary testing and counseling |
| CTC | Muhimbii—Care and Treatment Center |

## Appendix A

**Table A1.** Search strategy employed to identify studies in the systematic review. Methods are detailed in Section 2.2 of the manuscript.

| PubMed | | |
|---|---|---|
| **Set** | **Search terms** | **Results** |
| #1 | "HIV Infections" [Mesh] OR "HIV" [Mesh] OR "HIV Long-Term Survivors" [Mesh] OR "HIV Testing" [Mesh] OR "HIV Seroprevalence" [Mesh] OR HIV [tiab] OR AIDS [tiab] OR "Human immunodeficiency virus" [tiab] OR "acquired immunodeficiency syndrome" [tiab] | 474,722 |
| #2 | prevalence [Mesh] OR Incidence [Mesh] OR prevalence [tiab] OR prevalent [tiab] OR incidence [tiab] OR statistics and numerical data [sh] OR epidemiology [sh] OR statistics [tiab] OR rate [tiab] OR rates [tiab] OR population [tiab] OR seroprevalence [tiab] OR epidemiology [tiab] | 7,186,324 |
| #3 | "Tanzania" [Mesh] OR Tanzania [all fields] OR Kilimanjaro [all fields] OR Moshi [all fields] OR Tanzanian [all fields] OR Kili [all fields] | 21,480 |
| #4 | #1 AND #2 AND #3 | 2788 |
| #5 | #4 AND ("2015/01/01" [Date Publication]: "3000" [Date Publication]) | 1155 |
| Embase | | |
| **Set** | **Search terms** | **Results** |
| #1 | 'Human immunodeficiency virus infection'/exp OR 'Human immunodeficiency virus'/exp OR 'Human immunodeficiency virus infected patient'/exp OR 'HIV test'/exp OR 'Human immunodeficiency virus prevalence'/exp OR HIV: ab, ti OR AIDS: ab, ti OR "Human immunodeficiency virus": ab, ti OR "acquired immunodeficiency syndrome": ab, ti | 647,800 |
| #2 | 'prevalence'/exp OR 'incidence'/exp OR 'human immunodeficiency virus infection'/exp/dm_ep OR prevalence: ab, ti OR prevalent: ab, ti OR incidence: ab, ti OR statistics: ab, ti OR rate: ab, ti OR rates: ab, ti OR population: ab, ti OR seroprevalence: ab, ti OR epidemiology: ab, ti | 7,903,764 |
| #3 | 'Tanzania'/exp OR Tanzania: ab, ti, ca or Kilimanjaro: ab, ti, ca or Moshi: ab, ti, ca OR Tanzanian: ab, ti, ca OR Kili: ab, ti, ca | 22,246 |

**Table A1.** *Cont.*

| #4 | #1 AND #2 AND #3 | 2519 |
|----|------------------|------|
| #5 | #4 AND [2015-2022]/py | 1073 |

| **Scopus** | | |
|---|---|---|
| **Set** | **Search terms** | **Results** |
| #1 | TITLE-ABS-KEY (HIV OR AIDS OR "Human immunodeficiency virus" OR "acquired immunodeficiency syndrome") | 684,566 |
| #2 | TITLE-ABS-KEY (prevalence OR prevalent OR incidence OR statistics OR rate OR rates OR population OR seroprevalence OR epidemiology) | 13,234,711 |
| #3 | TITLE-ABS-KEY (Tanzania OR Kilimanjaro OR Moshi OR Tanzanian OR Kili) OR AFFILCOUNTRY (Tanzania) OR AFFILCITY (Moshi) | 49,412 |
| #4 | #1 AND #2 AND #3 | 3012 |
| #5 | #4 AND 2015–present | 1240 |

| **African Index Medicus** | | |
|---|---|---|
| **Set** | **Search terms** | **Results** |
| #1 | (HIV OR AIDS OR "Human immunodeficiency virus" OR "acquired immunodeficiency syndrome") AND (prevalence OR prevalent OR incidence OR statistics OR rate OR rates OR population OR seroprevalence OR epidemiology) AND (Tanzania OR Kilimanjaro OR Moshi OR Tanzanian OR Kili); 2015-2022 | 2 |

**Appendix B**

Questions used to assess the quality of included studies. Methods are detailed in Section 2.2.4, results in Section 3.2.1, and conclusions in Section 5.2.

**Quality Assessment Questions**

1. Was the research question or objective in this paper clearly stated?
2. Did the authors use an appropriate method to answer their question?
3. Was the study population clearly specified and defined?
4. Were the cases recruited in an acceptable way?
5. Were all subjects selected or recruited from the same or similar populations (including the same time period)?
6. Were inclusion and exclusion criteria for being in the study prespecified and applied uniformly to all participants?
7. Was a sample size justification, power description, or variance and effect estimates provided?
8. Were the outcome measures (dependent variables) clearly defined, valid, reliable, and implemented consistently across all study participants?

**Appendix C**

**Table A2.** Quality assessment of included studies organized by clinical and nonclinical population groups. Quality results are discussed in Sections 3.2.1 and 5.2.

| Population Group | Study | Quality Rating (#/8) | Outcome Type |
|---|---|---|---|
| **Clinical Populations** | | | |
| **Pregnant women** | Gamell, 2017 [43] | 7 | primary |
| | Konje, 2018 [53] | 7 | primary |
| | Chibwe, 2019 [83] | 8 | secondary |
| | Ng'wamkai, 2019 [67] | 8 | primary |
| **Pregnant women attending antenatal care clinics (ANC)** | Manyahi, 2017 [58] | 8 | primary |

**Table A2.** *Cont.*

| Population Group | Study | Quality Rating (#/8) | Outcome Type |
|---|---|---|---|
| Pregnant women at delivery | Lawi, 2015 [54] | 8 | primary |
| Women who delivered within the last 2 years | Adinan, 2019 [36] | 8 | primary |
| Newly delivered mothers | Nungu, 2019 [71] | 8 | primary |
| Women screened for cervical cancer | Chambuso, 2016 [39] | 7 | primary |
| | Mchome, 2020 [84] | 7 | secondary |
| | Chinn, 2021 [85] | 8 | secondary |
| | Katanga, 2021 [86] | 7 | secondary |
| Women with cervical cancer | Lovgren, 2016 [56] | 7 | primary |
| | Khamis, 2021 [87] | 7 | secondary |
| Men undergoing voluntary circumcision | Bazant, 2016 [88] | 8 | secondary |
| Febrile adult patients | Boillat-Blanco, 2018 [89] | 7 | secondary |
| Patients with Fournier's gangrene | Chalya, 2015 [90] | 7 | secondary |
| Patients who were managed for domestic-violence-related trauma | Chalya, 2015 [91] | 7 | secondary |
| Hospital patients | Kilale, 2016 [92] | 7 | secondary |
| Patients in outpatient department clinics | Cham, 2019 [35] | 7 | primary |
| Multiple injury patients | Issa, 2018 [93] | 8 | secondary |
| ED patients with injury | Hyuha, 2021 [45] | 8 | primary |
| Confirmed TB patients | Denti, 2015 [79] | 8 | secondary |
| | Gunda, 2017 [44] | 7 | primary |
| | Friis, 2018 [42] | 7 | primary |
| | Kidenya, 2018 [80] | 7 | secondary |
| | Mhimbira, 2019 [81] | 7 | secondary |
| Patients on treatment for TB | Munseri, 2019 [94] | 8 | secondary |
| Suspected TB patients | Hoza, 2016 [95] | 7 | secondary |
| | Reither, 2015 [96] | 7 | secondary |
| Presumptive pulmonary TB patients | Mhimbira, 2015 [77] | 7 | secondary |
| Patients with bacteriologically confirmed pulmonary TB | Senkoro, 2016 [97] | 8 | secondary |
| Patients who completed 20–24 weeks of TB treatment | Manji, 2016 [98] | 7 | secondary |
| Patients treated with levofloxacin as part of MDR-TB regimen | Mohamed, 2021 [34] | 7 | secondary |
| Adults who received TB treatment within 2 years | Mpagama, 2021 [99] | 8 | secondary |
| Head and neck cancer patients | Gilyoma, 2015 [100] | 7 | secondary |
| PWID attending methadone clinics | Lambdin, 2017 [101] | 7 | secondary |
| | Kilonzo, 2021 [52] | 7 | primary |

**Table A2.** *Cont.*

| Population Group | Study | Quality Rating (#/8) | Outcome Type |
|---|---|---|---|
| HCV-seropositive patients enrolled in the local opioid substitution treatment center | Mohamed, 2017 [102] | 7 | secondary |
| Patients with a neurological disorder admitted to the medical ward | Laizer, 2019 [103] | 7 | secondary |
| Blood donors | Lidenge, 2020 [104] | 7 | secondary |
| | Mremi, 2021 [82] | 7 | secondary |
| Patients with first stroke | Matuja, 2020 [105] | 7 | secondary |
| Patients with chronic kidney disease (CKD) | Meremo, 2018 [106] | 7 | secondary |
| Women with and without *S. mansoni* infection | Mishra, 2019 [107] | 7 | secondary |
| Patients with surgical acute abdomen | Sravanam, 2018 [74] | 7 | primary |
| VCT clients, ANC attendees, blood Donors, and CTC patients | Urio, 2015 [76] | 7 | primary |
| **Nonclinical populations** | | | |
| Cohabiting couples | Ngilangwa, 2015 [68] | 7 | primary |
| Married and cohabitating heterosexual adults | Mtenga, 2015 [64] | 7 | primary |
| Adults over 50 | Senkoro, 2016 [78] | 8 | secondary |
| | Mtowa, 2017 [65] | 7 | primary |
| | Swai, 2017 [75] | 7 | primary |
| | Muiruri, 2019 [66] | 7 | primary |
| Adult women | Faber, 2017 [41] | 7 | primary |
| | Hjort, 2019 [109] | 8 | secondary |
| | Baldur-Felskov, 2019 [110] | 7 | secondary |
| | Mchome, 2021 [60] | 7 | primary |
| Sexually active women | Safari, 2019 [111] | 7 | secondary |
| Female bar workers | Barnhart, 2019 [38] | 7 | primary |
| Adult men | Norris, 2017 [69] | 7 | primary |
| | Olesen, 2017 [112] | 7 | secondary |
| Men living in rural Tanzania | Downs, 2017 [40] | 7 | primary |
| Men who have sex with men (MSM) | Ahaneku, 2016 [37] | 8 | primary |
| | Ishungisa, 2020 [46] | 8 | primary |
| | Khatib, 2017 [51] | 7 | primary |
| | Mmbaga, 2017 [63] | 8 | primary |
| | Mmbaga, 2020 [113] | 8 | primary |
| | Mizinduko, 2020 [61] | 8 | primary |
| Male plantation residents | Norris 2017 [70] | 7 | primary |
| Fisherfolk | Kapesa, 2018 [47] | 8 | primary |
| | Kapiga, 2021 [48] | 7 | primary |
| | Panga, 2021 [72] | 8 | primary |

**Table A2.** *Cont.*

| Population Group | Study | Quality Rating (#/8) | Outcome Type |
|---|---|---|---|
| **People who inject drugs (PWID)** | Matiko, 2015 [59] | 8 | primary |
| | Khatib, 2017 [50] | 7 | primary |
| | Mmbaga, 2017 [62] | 8 | primary |
| | Leyna, 2019 [55] | 8 | primary |
| | Kawambwa, 2020 [49] | 8 | primary |
| | Minja, 2021 [108] | 8 | secondary |

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
