# Peer review of "HIV Prevalence among Injury Patients Compared to Other High-Risk Groups in Tanzania"

_traumacare, doi:10.3390/traumacare2030041_

Round 1

Reviewer 1 Report

Dear Authors,

I think your work has successfully merged the collection of primary data with a scoping review. Kudos.

If I have a criticism, it would be that you should write your future articles with more emphasis on detailing the hypotheses of your manuscript.

Strengths of the paper

1.      Interesting foundational academic argument for the paper.

2.      Strength of data is good.

3.      Well written – no English Language corrections needed.

Critical points to note:

1.      In part 1, the authors state that they hypothesize that “injury patients will have a higher………..” This is good, but could the authors please state this in numericized form? For example: H1: ED patients would have higher prevalence for HIV than non ED admitted patients in hospitals in Nigeria. Please do consider this.

2.      In Section 2 “Materials and Methods”, the authors should explain the justification used for both of these research methods. For example, why did the authors used the STROBE guidelines as opposed to any other guidelines? This would strengthen the understanding of the reader of the methodology. Additionally, the authors should explain why a systematic review, as opposed to a scoping or literature review, was taken as part of the methodology. This again would strengthen the basic foundation of the manuscript and benefit the reader.

3.      Limitations are well thought out – but perhaps the authors may want to note that the limitations should also explicitly state that the articles used for the systematic review are limited to Tanzania and no other country. This may skew the results, i.e. already have a bias towards the hypothesis of the authors.

Thank you and best wishes.

Reviewer 3 Report

The manuscript is very interesting. Although the conclusions seem to go too far. Perhaps it is not about the fact of participating in the accident itself, but about the financial / social status related to the possession of a means of transport (these are mainly young men).

The advantage of work are the diagrams that facilitate the understanding of the collected data. The value of the work is increased by the review part.

Unfortunately, even the simplest statistical analysis is definitely lacking in the work, which is why I recommend major revision.

Round 2

Reviewer 3 Report

I am satisfied with the authors' answers, although I regret that they have not decided to analyze their data in such a way that the obtained results are more transparent.